*Correspondence*

EMBO
Molecular Medicine

# High levels of circulating osteopontin in inflammatory lung disease regardless of Sars-CoV-2 infection

Giuseppe Cappellano[1,2,*] , Hugo Abreu[1,2], Davide Raineri[1,2], Lorenza Scotti[3], Luigi Castello[3,4], Rosanna Vaschetto[3] & Annalisa Chiocchetti[1,2]

Gibellini *et al* (2020) showed that the bioenergetic alteration of the monocyte compartment affects the inflammatory response to Sars-CoV-2 infection. To investigate the role of circulating monocytes in COVID-19-related inflammation, the authors compared expression profiles of molecules involved in monocyte recruitment and migration, in the plasma of healthy controls (HC) and Sars-CoV-2[+] patients. They found a significant increase of osteopontin (OPN) levels in Sars-CoV-2[+] patients.

OPN is a multifaceted molecule involved in the inflammatory response: it modulates leukocyte activation, migration, and differentiation and induces cytokine secretion in both acute and chronic inflammation (Clemente *et al*, 2016).

Since 2004, the present group has focused on elucidating the role of OPN in a broad array of pathological conditions. Increased OPN levels were found in patients with autoimmune lymphoproliferative syndrome (Chiocchetti *et al*, 2004), Alzheimer's disease (Comi *et al*, 2010), and recently in sepsis (Castello *et al*, 2019). ICOS-L has also recently been identified as a new receptor of OPN, shifting the attention from integrins and CD44 (the previously known ligands of OPN) toward this new receptor (Raineri *et al*, 2020). ICOS-L is expressed by antigen-presenting cells, comprising monocytes, and may sustain their activation/differentiation. It

was thus attempted to investigate whether OPN might also be related to COVID-19, by focusing on its levels in circulation, with the aim of establishing a new and easily accessible biomarker in blood.

From April 2020 to December 2020, 52 hospitalized patients with confirmed Sars-CoV-2 infection were enrolled at Maggiore della Carità University Hospital in Novara, Piedmont, a region considered to be one of the Italian epicenters of the pandemic (Cappellano *et al*, 2021). Patients were hospitalized for acute respiratory or influenza-like symptoms during the two waves of Sars-CoV-2 infection and stratified by the results of molecular analysis on nasal swabs. OPN levels were measured by ELISA (R&D System) and, in agreement with Gibellini *et al*, OPN levels were found to be increased in Sars-CoV-2 patients compared to non-comorbid and non-hospitalized healthy controls (HC; Figure 1).

However, OPN is reportedly associated with a diverse spectrum of lung and other diseases, and increased levels have been found in patients with interstitial (Kadota *et al*, 2005) and eosinophilic pneumonias (Ueno *et al*, 2010). It was thus decided to measure OPN levels also in a cohort of Sars-CoV-2-negative patients hospitalized with pneumonia-associated symptoms ($n = 56$). Diagnosis at hospital admission was respiratory failure related to acute or acute-on-chronic heart failure ($n = 17$), sepsis ($n = 16$ of which 25% due to community-acquired

**Figure 1. OPN plasma levels are increased in both Sars-CoV-2[+] and Sars-CoV-2[−] patients.**

OPN plasma levels were determined in 52 Sars-CoV-2[+] patients, 56 Sars-CoV-2[−] patients, and 54 HC, at the end of the study period in the stored samples collected at the time of enrollment, using commercially available ELISA (enzyme-linked immunosorbent assay; R&D system) following the manufacturer's protocol. Informed consent was obtained from all human subjects or their legal representatives involved in the study and approved by the local ethics committee (CE67/20). The experiments conformed to the principles set out in the WMA Declaration of Helsinki and the Department of Health and Human Services Belmont Report. Data are shown as mean ± SEM. Data normality was assessed using D'Agostino–Pearson test; the non-parametric Kruskal–Wallis test with Dunn's correction was then used (****$P < 0.001$ vs. HC).

pneumonia), intracranial bleeding ($n = 8$), trauma ($n = 5$), and others ($n = 10$). OPN plasma levels were found to be increased in

1   Department of Health Sciences, Interdisciplinary Research Center of Autoimmune Diseases-IRCAD, Università del Piemonte Orientale, Novara, Italy
2   Center for Translational Research on Autoimmune and Allergic Disease—CAAD, Università del Piemonte Orientale, Novara, Italy
3   Department of Translational Medicine, Universita' del Piemonte Orientale, Novara, Italy
4   Emergency Department, "Maggiore della Carità" University Hospital, Novara, Italy
    *Corresponding author. E-mail: giuseppe.cappellano@med.uniupo.it

DOI 10.15252/emmm.202114124 | EMBO Mol Med (2021) 13: e14124 | Published online 31 March 2021

Sars-CoV-2⁻ patients compared to HC, whereas no difference was found between Sars-CoV-2⁺ and Sars-CoV-2⁻ patients (Figure 1). These findings suggest that OPN cannot be used as a biomarker to discriminate Sars-CoV-2 infection at the current state of knowledge.

Inflammatory lung diseases may resolve or else progress to fibrosis, which is associated with increased OPN expression, since OPN modulates collagen fibrillogenesis and remodeling (Pardo *et al*, 2005). Post-inflammatory COVID-19 pulmonary fibrosis is emerging as a complication of Sars-CoV-2 infection, affecting around one-third of hospitalized Sars-CoV-2⁺ patients (Tale *et al*, 2020). Although no difference was found in OPN levels between Sars-CoV-2⁺ and Sars-CoV-2⁻ patients, it cannot be ruled out that OPN might play a pivotal role in post-Sars-CoV-2 infection. As Gibellini *et al*, (2020) infer, increased levels of OPN, along with other inflammatory cytokines, may contribute to COVID-19 immunopathogenesis, namely regarding the role of monocytes in inflammatory lung diseases. OPN deregulation may impact on COVID-19 in different ways: it might be associated either with disease severity or with the development of a fibrotic phenotype. Several factors in Sars-CoV-2 infection may influence OPN secretion, i.e., amount of inflammatory host response, disease progression, and therapy prescribed to the patients, i.e., anti-inflammatory drugs, corticosteroids, and heparin. Noteworthy, in the past we showed that anti-OPN autoantibodies, with neutralizing properties, are spontaneously produced during inflammatory conditions boosting OPN production (Clemente *et al*, 2017). Differential production of such response may impact on the level of resolution of the inflammatory burst. Furthermore, due to the multifaceted nature, OPN may also act as a pro-fibrotic protein. Currently, no fully proven options are available for the treatment of post-inflammatory COVID-19 pulmonary fibrosis, since the pathogenetic mechanism is not fully understood. Risk factors include age, history of smoking, severe illness, and prolonged mechanical ventilation, but monitoring OPN (and its autoantibodies) levels at discharge and during COVID-19 follow-up may be relevant, in view of considering OPN as a potential predictor factor of pulmonary fibrosis.

A well-designed study, considering all these variables, is needed to elucidate the association of OPN with disease outcome.

## Acknowledgements

This study was funded by the Italian Ministry of Education, University and Research (MIUR) program "Departments of Excellence 2018–2022", FOHN to AC and AGING Projects to AC and RV, Fondazione Cariplo 2019-3277 to AC, and European Union's Horizon 2020 Research and Innovation Program under Grant Agreement No. 953121—project FLAMIN-GO to AC.

## Author contributions

GC performed the experiments and wrote the manuscript. HA performed the experiments. DR analyzed the clinical database. LS performed the statistical analysis. LC and RS provide the biological samples and clinical data. AC conceived the study and wrote the manuscript. All authors reviewed, revised, and approved the final manuscript.

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
