## [Review Process File · EMBO Molecular Medicine]

High levels of circulating osteopontin in inflammatory lung disease regardless of Sars-Cov-2 infection.

Giuseppe Cappellano, Hugo Abreu, Davide Raineri, Lorenza Scotti, Luigi Castello, Rosanna Vaschetto, Annalisa Chiocchetti
DOI: [10.15252/emmm.202114124](https://doi.org/10.15252/emmm.202114124)

Corresponding author: Giuseppe Cappellano (giuseppe.cappellano@med.uniupo.it)

Review Timeline:

Submission Date:	23rd Feb 21
Editorial Decision:	9th Mar 21
Revision Received:	15th Mar 21
Accepted:	17th Mar 21

Editor: Zeljko Durdevic

Transaction Report:

9th Mar 2021

Dear Dr. Cappellano,

Thank you for the submission of your manuscript to EMBO Molecular Medicine. I am pleased to inform you that we will be able to accept your manuscript pending the following final amendments:

1) Please address all referee's comments.

2) Figures: Please upload individual, high-resolution figure file and name it Figure 1 also in the main text. Figure legend should remain at the end of the main manuscript text. For more information on figure presentation please check "Author Guidelines".

<https://www.embopress.org/page/journal/17574684/authorguide#datapresentationformat>

3) In the main manuscript file, please do the following:

- Add up to 5 keywords.

- Add contributions for all authors.

- Conflict of interest statement should be named "Conflict of interest".

- Include a statement that informed consent was obtained from all human subjects and that, in addition to the WMA Declaration of Helsinki, the experiments conformed to the principles set out in the Department of Health and Human Services Belmont Report.

4) Funding: Please make sure that information about all sources of funding are complete in both our submission system and in the manuscript.

5) As part of the EMBO Publications transparent editorial process initiative (see our Editorial at <http://embomolmed.embopress.org/content/2/9/329>), EMBO Molecular Medicine will publish online a Review Process File (RPF) to accompany accepted manuscripts. This file will be published in conjunction with your paper and will include the anonymous referee reports, your point-by-point response and all pertinent correspondence relating to the manuscript. Let us know whether you agree with the publication of the RPF and as here, if you want to remove or not any figures from it prior to publication. Please note that the Authors checklist will be published at the end of the RPF.

6) Please provide a point-by-point letter INCLUDING my comments as well as the reviewer's reports and your detailed responses (as Word file).

I look forward to reading a new revised version of your manuscript as soon as possible.

Yours sincerely,

Zeljko Durdevic

***** Reviewer's comments *****

Referee #1 (Remarks for Author):

In their correspondence "High levels of circulating osteopontin in inflammatory lung disease regardless of Sars-Cov-2 infection." the authors test osteopontin as a biomarker for SARS-CoV-2 infection. The rationale is based on a report by Gibellini et al. published in 2020 addressing monocyte alterations in SARS-CoV-2 and showing increased osteopontin levels in COVID-19 patients.

While Gibellini et al. had compared COVID-19 patients to healthy controls, the authors of this manuscript test whether increased osteopontin is specific for SARS-CoV-2 infection as opposed to other causes of pneumonia, which indeed turned out not to be the case. Even though this might not necessarily be surprising, it is an interesting finding worth reporting. Not least does it demonstrate that findings made with respect to COVID-19 need to be put into context and it needs to be discriminated between what is more general infection-associated inflammation and what is SARS-CoV-2-specific. There are some points to be addressed:

- 1) As an introduction it is stated that "Gibellini et al. (2020) showed that the bioenergetic alteration of the monocyte compartment affects the pathogenic response to Sars-Cov-2 infection". As such effects on the pathogenic response were not explicitly shown by Gibellini et al. and they rather demonstrated alterations to the monocytic compartment in COVID-19, please revise this introductory statement to more precisely reflect the findings of Gibellini et al.
- 2) "OPN is a multifaceted molecule involved in the inflammatory response: it modulates leukocyte activation, migration and differentiation and induces cytokine secretion both in acute and chronic inflammation.". Please provide a reference for the effects of OPN.
- 3) As OPN is affected in a broad array of conditions, the authors might want to indicate their criteria defining healthy controls.
- 4) Regarding the cohort of Sars-Cov-2 negative patients if possible please indicate whether these were viral, bacterial, any other pneumonia patients or how these characteristics were distributed within the cohort.
- 5) Even though OPN is not a marker for SARS-CoV-2 infection, could the authors speculate on whether increased OPN might be associated with a certain form or severity of COVID-19. There is quite some heterogeneity in the OPN levels reported in SARS-CoV-2 positive patients. It would be interesting to learn whether e.g. explicitly high levels are associated with severe disease. This might make OPN suitable not as a biomarker for SARS-CoV-2 per se but e.g. for the course of the acute disease or the intensity of inflammation.

Response to the reviewer

Referee #1 (Remarks for Author):

In their correspondence "High levels of circulating osteopontin in inflammatory lung disease regardless of Sars-Cov-2 infection." the authors test osteopontin as a biomarker for SARS-CoV-2 infection. The rationale is based on a report by Gibellini et al. published in 2020 addressing monocyte alterations in SARS-CoV-2 and showing increased osteopontin levels in COVID-19 patients.

While Gibellini et al. had compared COVID-19 patients to healthy controls, the authors of this manuscript test whether increased osteopontin is specific for SARS-CoV-2 infection as opposed to other causes of pneumonia, which indeed turned out not to be the case. Even though this might not necessarily be surprising, it is an interesting finding worth reporting. Not least does it demonstrate that findings made with respect to COVID-19 need to be put into context and it needs to be discriminated between what is more general infection-associated inflammation and what is SARS-CoV-2-specific. There are some points to be addressed:

Q1) As an introduction it is stated that "Gibellini et al. (2020) showed that the bioenergetic alteration of the monocyte compartment affects the pathogenic response to Sars-Cov-2 infection". As such effects on the pathogenic response were not explicitly shown by Gibellini et al. and they rather demonstrated alterations to the monocytic compartment in COVID-19, please revise this introductory statement to more precisely reflect the findings of Gibellini et al.

R1. We thank the reviewer for his/her comment, we revised the sentence, as following *"Gibellini et al. (2020) showed that the bioenergetic alteration of the monocyte compartment affects the inflammatory response to Sars-Cov-2 infection"*.

Q2) "OPN is a multifaceted molecule involved in the inflammatory response: it modulates leukocyte activation, migration and differentiation and induces cytokine secretion both in acute and chronic inflammation.". Please provide a reference for the effects of OPN.

R2. As suggested, by the reviewer, we added the missing reference (Clemente et al., 2016).

Q3) As OPN is affected in a broad array of conditions, the authors might want to indicate their criteria defining healthy controls.

R3. We thank the reviewer for his/her valuable suggestion. We edited the text to clarify it.

Q4) Regarding the cohort of Sars-Cov-2 negative patients if possible please indicate whether these were viral, bacterial, any other pneumonia patients or how these characteristics were distributed within the cohort.

R4. We thank the reviewer for his/her comment. We added the following sentences: *'Diagnosis at hospital admission were respiratory failure related to acute or acute-on-chronic heart failure (n=17), sepsis (n=16 of which 25% due to community acquired pneumonia), intracranial bleeding (n=8), trauma (n=5) and others (n=10)'*.

Q5) Even though OPN is not a marker for SARS-CoV-2 infection, could the authors speculate on whether increased OPN might be associated with a certain form or severity of COVID-19. There is quite some heterogeneity in the OPN levels reported in SARS-CoV-2 positive patients. It would be interesting to learn whether e.g. explicitly high levels are associated with severe disease. This might make OPN suitable not as a biomarker for SARS-CoV-2 per se but e.g. for the course of the acute disease or the intensity of inflammation.

R5. The reviewer is right; we added in the text the following sentences: *"OPN deregulation may impact on COVID-19 in different ways: it might either be associated with disease severity or with*

the development of a fibrotic phenotype. Several factors in Sars-Cov-2 infection may influence OPN secretion i.e., amount of inflammatory host response, disease progression and therapy prescribed to the patients i.e., anti-inflammatory drugs, corticosteroids and heparin. Noteworthy, in the past we showed that anti-OPN autoantibodies, with neutralizing properties, are spontaneously produced during inflammatory conditions boosting OPN production (Clemente et al., 2017). Differential production of such response, may impact on the level of resolution of the inflammatory burst. Furthermore, due to the multifaceted nature, OPN may also act as a pro-fibrotic protein. Currently, no fully proven options are available for the treatment of post-inflammatory COVID-19 pulmonary fibrosis, since the pathogenetic mechanism is not fully understood. Risk factors include age, history of smoking, severe illness, and prolonged mechanical ventilation, but monitoring OPN (and its autoantibodies) levels at discharge and during COVID-19 follow-up, may be relevant, in view of considering OPN as a potential predictor factor of pulmonary fibrosis. A well-designed study, considering all these variables, is needed to elucidate the association of OPN with disease outcome''.

We are pleased to inform you that your manuscript is accepted for publication and is now being sent to our publisher to be included in the next available issue of EMBO Molecular Medicine.

Corresponding Author Name: GIUSEPPE CAPPELLANO

Journal Submitted to: EMBO MOLECULAR MEDICINE

Manuscript Number: EMM-2021-14124